# Targeted Mass Spectrometry Reveals Interferon-Dependent Eicosanoid and Fatty Acid Alterations in Chronic Myeloid Leukaemia

**DOI:** 10.3390/ijms242115513

**Published:** 2023-10-24

**Authors:** Hannah C. Scott, Simeon D. Draganov, Zhanru Yu, Benedikt M. Kessler, Adán Pinto-Fernández

**Affiliations:** 1Chinese Academy of Medical Sciences Oxford Institute, Nuffield Department of Medicine, University of Oxford, Oxford OX3 7BN, UK; simeon.draganov@ndm.ox.ac.uk (S.D.D.); zhanru.yu@ndm.ox.ac.uk (Z.Y.); benedikt.kessler@ndm.ox.ac.uk (B.M.K.); 2Target Discovery Institute, Centre for Medicines Discovery, Nuffield Department of Medicine, University of Oxford, Oxford OX3 7FZ, UK

**Keywords:** bioactive lipids, eicosanoids, fatty acids, mass spectrometry, lipidomics, innate immunity, type I interferon response, chronic myeloid leukaemia, cancer inflammation, cancer metabolism

## Abstract

Bioactive lipids are involved in cellular signalling events with links to human disease. Many of these are involved in inflammation under normal and pathological conditions. Despite being attractive molecules from a pharmacological point of view, the detection and quantification of lipids has been a major challenge. Here, we have optimised a liquid chromatography–dynamic multiple reaction monitoring–targeted mass spectrometry (LC-dMRM-MS) approach to profile eicosanoids and fatty acids in biological samples. In particular, by applying this analytic workflow to study a cellular model of chronic myeloid leukaemia (CML), we found that the levels of intra- and extracellular 2-Arachidonoylglycerol (2-AG), intracellular Arachidonic Acid (AA), extracellular Prostaglandin F_2α_ (PGF_2α_), extracellular 5-Hydroxyeicosatetraenoic acid (5-HETE), extracellular Palmitic acid (PA, C16:0) and extracellular Stearic acid (SA, C18:0), were altered in response to immunomodulation by type I interferon (IFN-I), a currently approved treatment for CML. Our observations indicate changes in eicosanoid and fatty acid metabolism, with potential relevance in the context of cancer inflammation and CML.

## 1. Introduction

Bioactive lipids regulate cellular functions and contribute to tissue homeostasis and pathology [1]. Four families of bioactive lipids are involved in inflammation and immune regulation: eicosanoids, specialised pro-resolving mediators, lysoglycerophospholipids/sphingolipids and endocannabinoids, which are generated from ω-3 or ω-6 polyunsaturated fatty acids (PUFAs). Eicosanoids are key elements in inflammation, and have been linked with diseases, such as viral infections, neurodegenerative disorders [2], rheumatoid arthritis [3], atherosclerosis [4,5], acute coronary syndrome [6], cancer [7,8], systemic lupus erythematosus [9], multiple sclerosis [10], liver injury [11], celiac disease [12], diabetes [13], cystic fibrosis [14], renovascular disease [15], asthma [16] and muscle dysfunction [17]. In addition, fatty acids, the common components of complex lipids, often act as immunomodulatory factors [18,19]. 

In general, lipids are challenging molecules to characterise due to the limited analytical capacity to identify, quantify and annotate them in biological samples, which has been an impediment to the advancement of discovering molecular mechanisms of disease, biomarker discovery and drug development. Mass spectrometry (MS) approaches, predominantly in combination with liquid chromatography (LC), are widely used for the analysis of biomolecules [20,21,22,23], including eicosanoids [24,25]. More recently, LC has been coupled with dynamic multiple reaction monitoring (dMRM) workflows for the targeted detection of specific subsets of eicosanoids [26,27], and high-resolution (MRMHR) methods to generate a library of high-resolution fragmentation spectra [28] have been developed. In addition, eicosanoid profiling by LC-MS has previously been performed on human plasma [29], clinical samples [30] and mouse models of pathophysiological states [31]. However, such analyses remain challenging due to limitations in the accurate assignment and detection of endogenous species [32,33]. One reason is a lack of thoroughness in optimising truly representative sample matrix variations, instead of using “surrogate matrices” (such as BSA in a sample buffer, which would be a matrix approximation) and applying more efficient extraction methodologies adapted to specific lipid subsets within specific sample types (e.g., cell and cell supernatant samples).

In this study, an intracellular and extracellular analysis (the media in which the respective cells are cultured) of eicosanoids and fatty acids was performed using a tailored extraction and dMRM method to maximise the sensitivity of the detection of endogenous species. Since bioactive lipids such as eicosanoids have important roles in inflammation and cancer, we performed a proof-of-concept experiment to study the effect of the pro-inflammatory cytokine interferon α 2 (IFNα-2b, hereafter IFN-I) on HAP1 cells. HAP1 cells were derived from a chronic myeloid leukaemia (CML) patient and have been extensively used in different translational studies [34,35,36]. IFN-I is a cytokine secreted by most cells in response to viral infection, and has known immunostimulatory and anticancer properties [37]; specifically, the IFN-I subtype IFNα2 has been approved, alone and in combination with tyrosine kinase inhibitors (TKIs), for the treatment of CML [38]. 

Here, due to the optimisation of a novel LC-dMRM-MS targeted method suitable for the analysis of a selection of eicosanoids and fatty acids in cells and supernatants, we were able to uncover changes in the lipidome of CML samples induced by a clinically relevant treatment, IFN-I. These lipids were selected for their known roles in inflammation and immune signalling, especially within the context of cancer and CML. For the subset of eicosanoids, we endeavoured to include representatives of as many different types as possible, such as the 5-HETE family, Leukotrienes and Prostaglandins. The selection process was also dependent on standard availability and compatibility with LC/MS analytical conditions such as chromatographic separation and ionisation efficiency. 

Of particular interest, we found that IFN-I affected intra- and extracellular concentrations of 2-Arachidonoylglycerol (2-AG), the intracellular concentration of Arachidonic Acid (AA) and the extracellular concentrations of Prostaglandin F_2α_ (PGF_2α_), 5-Hydroxyeicosatetraenoic acid (5-HETE), Palmitic acid (PA, C16:0) and Stearic acid (SA, C18:0), suggesting a role of these lipids in the inflammatory state of CML tumours that has not yet been described.

## 2. Results

A representative group of eicosanoids and fatty acids (Figure 1a) was selected based on their inflammatory roles and potential regulation by IFN-I. We focused on improving the identification and separation of lipid standards (chromatographic separation), their ionisation (source parameters) and their quantitation by developing and implementing a dMRM method. We also applied this method to biological samples, where we optimised the amount of starting material, the lipid extraction method, and determined the specific matrix effects upon each lipid in two sample types (cells and cell supernatant), as shown in Figure 1b.

### 2.1. Liquid Chromatography–Dynamic Multiple Reaction Monitoring–Mass Spectrometry (LC-dMRM-MS) Method Optimisation for Bioactive Lipid Standards

The analysis of lipid standards (Table 1) was first performed in MS^1^ mode to detect the corresponding precursor ions. Following tandem mass spectrometry (MS/MS; MS^2^), an extracted ion chromatogram of the most intense fragment ion was used for lipid assignment. The MS^2^ spectra were then compared to either acquired or predicted MS^2^ spectra in the Human Metabolome Database (HMDB) for confirmation of identity by using one to three fragment ions as a reference, as shown in Appendix A (Appendix A). Once discovered, the retention time was then used to assign the retention time window in the dMRM acquisition parameters.

Implementing dMRM increases the ion of interest’s response by removal of interfering signals (noise) and acquiring data on specific product ions only when the corresponding precursor ion is present. dMRM also improves dwell times during data acquisition by applying a retention time window that restricts when data are obtained for specific transitions, resulting in superior signal-to-noise (S/N) ratios. Non-overlapping ion transmissions also optimises dwell time and therefore sensitivity. dMRM transitions of lipid standards (Table 1) were identified using the Agilent MRM optimiser software (v. B.07.00), which applies a collisional energy (CE) ramp and suggests transitions based on the abundance and the mass-to-charge ratio (*m*/*z*) values. Manual checking of these transitions was performed, as the ion implied to have the highest abundance by the software (the quantitation ion, the “quant ion”) did not always align with the ion with the actual greatest response. By this method, we have identified up to four dMRM transitions that can be used as quant and qualifier ions (“qual ions”) (Table 1 and Appendix A). The “quant ion” is used for quantification, and the other three ions (“qual ions”) are used to corroborate identification. Analysis of lipids in the presence of ammonium acetate (which is our mobile phase modifier) can create ammonium adduct ions of lipids. In our methodology, an example of this phenomenon is the precursor ion of 5-HETE, which was an ammonium adduct with a *m*/*z* of 338.3. Structures of all lipid standards analysed and the corresponding proposed quant ion structures are shown in Appendix A.

The chromatographic separation of lipids reduces the complexity of biological samples and decreases matrix effects. The chromatogram, in Appendix A, displays the dMRM transitions for an equimolar ratio of all the lipids profiled. On the x-axis is the elution time (counts vs. acquisition time, also known as the retention time), and on the y-axis is the response (counts, AUC = area under curve). Not all responses are the same for all the lipids, even when at the same on-column concentration, as lipids are ionised at different efficiencies. Interestingly, it was not possible to chromatographically separate the regio-isomers species PGD_2_ and PGE_2_ reliably or sufficiently enough to determine their individual concentrations with this particular methodology. The results reported here are to be interpreted as the concentration of one, the other or both those metabolites combined. Otherwise, our method has sufficiently adequate separation, preventing overlapping MRMs, which results in longer scan times for each MRM, increasing the points per peak for each lipid. In the case of the co-eluting compounds, for example SA and PGF_2α_, compound-specific precursor and product ions allow for their accurate identification. 

We observed that the retention time for certain lipids could have a slight variation at different column loadings; therefore, the dMRM retention time window in the analysis method (used to trigger data acquisition for the respective transition at that particular time) was set wider for those lipids. In addition, peak shape can also change at different column loadings, with shoulders or tails on some peaks; therefore, we have included these in our analysis. For some lipids, lower on-column concentrations appeared to result in jagged, not smooth, peaks; therefore, it was necessary to add smoothing during data processing. To further improve the ionisation efficiency of lipids, all MS parameters were adjusted to an optimal value (Table 2). For example, we found that a source gas temperature of 80 °C was optimal for the identification of intact lipids by MS^2^, due to a lower degree of in-source fragmentation, allowing for the identification of fairly intact lipids. However, for dMRM analysis, the most optimal precursor-to-product ion responses were obtained at a source gas temperature of 280 °C. Once the optimal MS parameters were determined, we subsequently validated the robustness and reproducibility of our optimised method by intraday variation analysis, whereby lipid standards were combined and analysed in technical triplicate at two concentrations, 0.3 and 3 pmol on-column, respectively (Figure 2). The coefficient of variation (CV) was calculated between the mean of technical replicates at both concentrations. The observed CVs for all the lipids are lower than 20%, and lower than 5% for most lipids (six out of nine lipid standards), highlighting the repeatability of this method.

### 2.2. Bioactive Lipid Analysis in Biological Samples

To enhance the extraction of lipids from cells, three different liquid–liquid extraction (LLE) techniques were tested (Appendix A). The bioactive lipid mix (BLM, see Section 4.3. Standard Solutions), containing all the lipid standards of interest, was added to cells in the extraction buffer to yield a final concentration of 10 pmol/L on column. We established that Extraction 1 was the most optimal extraction method for PGD_2_, and Extraction 3 was optimal for extracting 2-AG, AA and LTC_4_. However, Extraction 2 was ideal for the majority of metabolites (5-HETE, 5-oxo-ETE, PA, PGF2α and SA) (Appendix A and Appendix A) and therefore it was the method of choice for the lipid extraction from cells in subsequent experiments.

Metabolite extraction from the media in which the cells were cultured (the cell supernatant) was performed in a similar manner to the metabolite extraction from cells, with the exception that steps 1 and 2 were combined (Appendix A). We determined that Extraction 1 was the most optimal extraction method for 2-AG and LTC_4,_ and Extraction 2 was most optimal for the extraction of 5-HETE, AA and 5-oxo-ETE (Appendix A and Appendix A). Extraction 3 resulted in the extraction of all lipids, with the highest extraction efficiency for four of the nine lipids analysed (PA, PGF_2α_, PGD_2_/E_2_ and SA), and therefore it was the method of choice for lipid extraction from cell supernatant samples in subsequent experiments. 

We subsequently determined the matrix effect for cell and cell supernatant samples (see Section 4.7. Data Analysis), to evaluate whether the complex biological mixture enhances or suppresses ionisation for each lipid, by calculating a corresponding matrix RF value (Appendix A), which ranged from 0.34 to 40.06 for cells and 0.07 to 4.81 for cell supernatants (Appendix A). The high degree of variation in matrix RF values for each lipid highlights the importance of determining a matrix RF for every individual lipid analysed, in each specific sample type.

### 2.3. Analysis of the Bioactive Lipidome after IFN-I Treatment in Cancer Cells

To validate our methodology, we decided to profile the nine lipids of interest in a CML-derived cell line, upon treatment with IFN-I. This treatment resulted in significant changes to the intracellular lipidome in cell samples (Figure 3a,b, Appendix A), as well as the extracellular lipidome in cell supernatant samples (Figure 3a,c, Appendix A). We observed a decrease in the intra- and extracellular concentrations of 2-AG upon IFN-I treatment, and a similar decrease was seen in intracellular concentrations of AA, which was in line with data published in literature [39]. Although we cannot confidently say whether this was due to a decrease in the synthesis of these lipids or an increase in their degradation, their increased conversion into downstream lipid metabolites upon IFN-I treatment is another plausible mechanism. An increase in the extracellular levels of PGF_2α_ was observed, and the intracellular concentrations of 5-HETE did not change upon IFN-I treatment. This suggests that 2-AG and AA may be used to generate 5-HETE in the presence of IFN-I. There was no significant change in LTC_4_ in cells or in the cell supernatant, indicating no major changes in the synthesis or usage of this lipid. Intracellular amounts of PA and SA were not affected. However, extracellular amounts of PA and SA increased significantly. This would suggest an increase in the synthesis and secretion of these fatty acids. 5-oxo-ETE, PGD_2_ and PGE_2_ were not identified in either cell or cell supernatant fractions. Nonetheless, this method can be used to confidently detect minute changes for all these lipids (even if the changes are not statistically significant). The overall change in the bioactive lipid profile after a 36 h IFN-I treatment is shown in Figure 3d. 

## 3. Discussion

The primary objective of this article was to optimise and implement a targeted lipidomics workflow in a disease-relevant cellular model to monitor the effects of an approved form of treatment. We have used HAP1 cells as a model for CML, as HAP1 cells are derived from CML patient cells and have the BCR-ABL gene, which is common in leukaemia’s, turning the myeloid cell into a chronic myeloid cell [34,35], and IFN-I (IFNα2) as a treatment. IFN-I is a currently used, alone and in combination with tyrosine kinase inhibitors (TKIs), form of treatment for CML in the clinics [38].

First, we have established the importance of investigating the appropriate extraction technique for the specific sample type (e.g., cells or cell supernatant). Second, we considered the matrix effect of every lipid analysed within the specific sample type. 

Due to intrinsic molecular differences, innate variation and organic solvent preferences during extraction, there is no agreement on the best sample preparation and analysis of lipids. As a consequence, analysing different subclasses simultaneously is challenging. In this study, we have optimised extraction, chromatography and MS parameters, which provide the best conditions for the identification and quantitation of a set of eicosanoids and fatty acids with well-established roles in inflammation. 

Solid-phase extraction (SPE) methods can be used to reduce the complexity, and to concentrate the samples, prior to analysis. However, we found that SPE is unreliable for concentrating a variety of lipids simultaneously, resulting in the loss of some metabolites (unpublished observations). Therefore, we opted to optimise LLE methods and concentrate samples subsequently by resuspension of the dried lipids into smaller volumes. Due to differences in the natural abundance of these metabolites, and the MS response, we were able to dilute the sample for those lipids with a higher natural concentration and/or a higher ion intensity (such as intracellular 2-AG). A separate method could be implemented for the analysis of 2-AG, LTC_4_ and PGF_2α_, as even at low on-column concentrations these lipids have a large AUC. For instance, the standards and samples could be diluted further than reported here. Pre-analytical influences may change lipid concentrations or matrices, which is why we prefer to use extensive matrix-dependent normalisations for each analyte within a “true” matrix (e.g., not a surrogate matrix), and demonstrate its application tailored to specific endogenous lipid (eicosanoids and fatty acids) detection in clinically relevant biological samples. However, to bypass normalisation calculations, to provide information on the regulatory network and to identify important metabolite conversions within the pathway, ratios between metabolites could be used instead (e.g., PGE_2_:PGF_2α_). In this analysis, there is variability in response curves for the lipids (R^2^ values from 0.950 to 0.999; Appendix A and Appendix A), and variation in the matrix effects (RF values from 0.34 to 40.06), highlighting the need to determine the responses for every lipid quantified, as the results would otherwise vary significantly from the observed amounts. Nonetheless, the reproducibility of detection for our method achieved technical CV values of less than 20% and were within 5–10% for most species (Figure 2). Our intraday variation for all lipids analysed was below 20% CV, with 5-oxo-ETE, SA, 2-AG, PA and 5-HETE being below 5%. 

Recommended options for the future development of this analysis could include chemical derivatisation of the lipids, exploring an analysis in negative mode electrospray ionisation (ESI−), using an ultrafast LC and a complementary proteomics analysis. Derivatization has been implemented for enhanced lipidomic analysis by MS [40], and it may boost the MS response for some lipids. However, it would need to be considered that derivatization itself is laborious and time-consuming, that different metabolites have differing derivatization efficiencies and that batch effects are common. It has also been described that lipids with carboxylic acid moieties will benefit from being analysed in negative polarity [41,42]. However, this may not be suitable for the simultaneous analysis of all the lipids reported here. For instance, and in line with our observations, ionisation of LTC_4_ has been already shown to be very efficient in positive polarity [43]. MS fast-switching between polarity modes could potentially further improve the sensitivity of detection. It is important to consider that PUFAs are easily oxidised and that this may lead to erroneous concentration reporting for some lipids. Using an ultrafast LC could reduce the total analysis time, increasing throughput and reducing the cost per sample and labour time (samples would not need to be added daily). In addition, sample quality could be improved as they would not need to be stored for as long in the autosampler. Integrating complementary proteomics data would lead to greater understanding and enhance confidence in the changes observed within the studied pathways. For instance, changes in the expression of some of the enzymes involved in the metabolism of eicosanoids and fatty acids have been associated with cancer development [44,45,46,47]. Pinpointing which enzymes are modified upon interferon treatment, using proteomics, may corroborate our findings and expand our knowledge of the effect of interferon treatment in CML patients.

The pro-inflammatory tumour microenvironment [37,48,49,50,51] and immune “hot” tumours show increased levels of inflammation and anticancer immune responses, such as T cell infiltration and the release of pro-inflammatory cytokines, including interferons. Defects in interferon signalling pathways during cancer treatment have been linked to immunotherapy resistance [52]. Our results suggest that IFN-I treatment significantly reduces the intracellular concentration of the eicosanoid precursors 2-AG and AA. However, the underlying molecular mechanism is as yet still unknown. We hypothesise that this reduction may be through decreased production of their precursors (2-AG-LPA or DAG), a decreased presence or activity of the respective enzymes which produce 2-AG-LAP and DAG (for example, 2-LPAP or DAGL) or by increased metabolism to downstream eicosanoids (such as PGF_2α_). The latter seems to be the case as both intracellular and extracellular concentrations of PGF_2α_ are increasing. This may suggest that cells treated with IFN-I are secreting excess PGF_2α_ out of the intracellular space. Increased concentrations of PGF_2α_ have been indicated as a driver of particular cancers [53,54,55], although its role in CML remains unclear. Secretion of SA and PA also appear increased upon IFN-I exposure, possibly through enhanced export. These fatty acids could contribute to the tumour microenvironment, act as key metabolites in reducing tumourigenicity and affect the interplay between tumour and immune cells. 5-oxo-ETE, PGD_2_ and PGE_2_ were not identified in the samples, indicating either their absence, that they fell below the lower limit of detection (LLOD, a signal-to-noise ratio of >3), and/or the matrix caused considerable ionisation suppression. Alternatively, perhaps there was unexpected in-source fragmentation, leading to differing parent or product ions from those that we monitored. 5-oxo-ETE could be rapidly metabolised to either 5-HETE or to Di-endoperoxidase [56,57], and therefore with the current treatment of 36 h it is very likely that the changes, if happening upon IFN-I treatment, will not be observed.

We were unable to chromatographically separate standards of the isobaric species PGD_2_ and PGE_2_. However, separation is possible with a method specifically dedicated to the analysis of these particular lipids. Employing normal-phase chiral chromatography, and/or alternative MS approaches (employing additional separation techniques, such as ion mobility), will isolate isobaric prostaglandins based on their chirality, mobility and/or collisional cross-section value. PGD_2_ can undergo dehydration to form Prostaglandin J_2_, and other Prostaglandin metabolites, with the dehydration of PGD_2_ being accelerated in the presence of serum albumin [58,59,60]. Since our tissue culture medium contains serum albumin, it might be contributing to the absence of PGD_2_ in our analysis. Previous studies have suggested that PGE_2_ is reduced in CML patients [61]; hence, this may be why our method was unable to detect PGE_2_ in samples. Upon IFN treatment, intracellular production of 5-HETE appears to be maintained, possibly through metabolism from AA, or by the reduction of 5-oxo-ETE (and NADP+) to 5-HETE (and NADPH) [57]. In addition, the extracellular quantity of 5-HETE significantly increases, which to our knowledge is novel biology and an area of CML research yet to be explored. Changes in LTC_4_ amount and activity, and LTC_4_ concentration, are cell-specific [62,63,64]. With CML researchers suggesting that there can be a steady state, an increase or a decrease in Leukotrienes in patients compared to healthy controls. Nonetheless, despite these disparities, inhibitors of Leukotriene signalling have reduced cancer growth in CML [65,66]. When these Leukotriene inhibitors have been used in combination with traditional methods to treat CML (tyrosine kinase inhibitors), the reduction in tumour growth was further increased [67]. Interestingly, our results, although not significant, show a slight decrease in intracellular LTC_4_ levels upon IFN-I stimulation. However, Leukotrienes can be readily metabolised [68], with LTC_4_ potentially being converted to Leukotriene D_4_; therefore, we cannot confirm that LTC_4_ concentrations are affected or not by 36 h of IFN-I treatment.

In conclusion, the present study describes an original, ultra-sensitive methodology, whereby LC and MS parameters were adapted to a specific set of lipids. Protocols for extracting these lipids from different sample types were modified considering the matrix effect of each individual lipid as well as the different sample types (cells and cell supernatant). We applied this workflow for an accurate and simultaneous analysis of a range of eicosanoids and fatty acids in a clinically relevant cellular model of CML. It would be interesting, in order to reflect the heterogeneity and diversity of CML patients, to analyse, using this analytical workflow, cells and cell supernatants from different CML cell lines, with different genetic backgrounds and, ideally, from primary cells representing the different subtypes of CML. This would include the lipidome analysis of patient samples with either the Philadelphia chromosome or the Bcr-Abl oncogene, at Chronic, Accelerated and Blast phases of CML. Our optimisation of sample preparation, technical parameters and data normalisation for different sample types, plus the application of dMRM transitions, maximises the sensitivity of detection, allowing detection at endogenous levels, even for very-low-abundance species. The observed improvements in detection and sensitivity are the hallmark of this analytical method, permitting us to resolve changes in relevant bioactive lipids in response to treatment with a pro-inflammatory cytokine in a cancer cell model. Translational application examples include the profiling of bioactive lipids in patient samples, for monitoring inflammation levels caused by pathology, or for treatment with immunomodulators (i.e., immunotherapy). In closing, we believe that this emphasises the benefits of using targeted MS in understanding pathophysiological states. 

## 4. Materials and Methods

### 4.1. Figures

Figure 1a,b, Figure 3d, Appendix A were generated in Biorender. 

### 4.2. Reagents 

Lipid standards: PGE_2_ (catalogue # sc-201225), PGF_2α_ (catalogue # sc-201227), 5-HETE (catalogue # sc-205136), 5-oxo-ETE (catalogue # sc-203783) and PGD_2_ (catalogue # sc-201221) were purchased from Santa Cruz Biotechnology (Dallas, TX, USA). SA (catalogue # 10011298–500 mg-CA), LTC_4_ (catalogue # 20210–25 ug-CAY) and PA (catalogue # 10006627-10 g-CAY) were acquired from Cayman Chemicals (Ann Arbor, MI, USA). AA (catalogue # ab120916) is from Abcam (Cambridge, UK). 2-AG (catalogue # A8973) was purchased from Sigma (St. Louis, MO, USA). LC-MS grade water (Catalogue # 115333) and MeOH (Catalogue # 106035) were available from Merck (Lowe, NJ, USA). Ammonium acetate (Catalogue # 10365260) and IPA (Catalogue # 15686670) were from Fisher Scientific (Waltham, MA, USA). MTBE was acquired from Acros organics (Catalogue # 3787 20010) and ACN was from Honeywell Riedel-de Haën (York, UK) (Catalogue # 348512.5L). IMDM was from Gibco (Billings, MT, USA) at Thermo Fisher Scientific (reagent # 12440053) and IFN-I was from Biotechne R&D systems (Minneapolis, MN, USA) (catalogue # 11105-1).

### 4.3. Standard Solutions

Stock concentrations of all lipid standards were prepared as individual aliquots at 100 µmol/L in 100% MeOH. These stocks were then pooled and serially diluted to a concentration of 100 nmol/L in 50% (*v*/*v*) MeOH to create an equimolar standard mixture, the bioactive lipid mix (BLM). The BLM was used as a spike, which was added to pooled samples (a fraction of all samples combined) to determine the matrix effects. 

### 4.4. Tissue Culture

HAP1 cells were cultured as previously described [69]. In brief, cells were cultured in IMDM media containing 10% FBS and grown until 60–70% confluence. Cells were then treated with 1000 U/mL of IFNα-2b for 36 h, until ~80% confluent. The surrounding medium (the cell supernatant) was collected and used for analysis of extracellular lipids. Both cell supernatant and cell plates were stored at −80 °C until lipid extraction.

### 4.5. Lipid Extraction

A simultaneous lysis and LLE was performed. For cells, the lysis was performed by scraping with 100% MeOH. The lysate was then added to 50% CHCl_3_ and 50% MBTE, which was then vortexed briefly and then spun at 12 rpm for 20 min at 4 °C. The sample was then centrifuged at 17× *g* for 5 min at 4 °C and the resulting organic fraction was removed and stored on ice. The remaining pellet underwent another round of vortexing, spinning and centrifuging in 50% CHCl_3_ and 50% MTBE, with the resulting organic fraction being pooled with the previous fraction. The final extraction step was performed with 50% MeOH and the sample was vortexed, spun and centrifuged again, with the organic fraction being pooled with the previous fractions. Once all organic fractions were collected for each sample, a small portion of each was removed and pooled together for a pooled sample. The samples and the pooled samples were then dried in a speedvac at 30 °C until dry and then stored at −20 °C until LC-MS analysis. 

To the dried lipid extracts 50% MeOH or the same volume of the BLM was added prior to analysis. The samples were briefly vortexed and then shaken at 400 rpm at 4 °C until homogenous. Cell supernatant samples were diluted 1 in 20. Cell samples were diluted 1 in 10 for analysis of AA, PGF_2α_, SA and 5-HETE or diluted 1 in 100 for analysis of 2-AG, PA and LTC_4_. Diluted samples were then transferred to LC-MS autosampler vials. 

### 4.6. LC-MS Method

Levels of 2-AG, AA, PGD_2_, PGE_2_, PGF_2α_, 5-HETE, 5-oxo-ETE, LTC_4_, PA and SA were analysed. Metabolites were quantified using an optimised dMRM method on a triple quadrupole mass spectrometer with a JetStream ESI source (Agilent 6490) coupled to a 1290 Agilent LC system.

Lipids were separated on an ACQUITY UPLC BEH C18 column (1.7 µm, 100 × 2.1 mm i.d., Waters) with mobile phase A of 2% IPA with 5 mM ammonium acetate and mobile phase B of 100% IPA with 5 mM ammonium acetate at 40 °C. The flow rate was set to 0.21 mL min^−1^ and the sample injection volume was 1 µL, 10 µL or 15 µL (depending on the response of the metabolites). The following gradient (% mobile phase B) was used: 0–1.5 min at 50% B, 1.5–9 min 70% B and 9–13 min 100% B. A wash with 100% mobile phase B and a wash with 100% mobile phase A were performed to clean the column before re-equilibration to starting conditions. The autosampler was maintained at 4 °C.

The following ESI+ source parameters were used: gas temp at 280 °C, gas flow 14 L/min, nebuliser at 20 psi, sheath gas temp at 250 °C, sheath gas flow at 11 L/min, capillary voltage 3000 V, nozzle voltage 1500 V, high-pressure RF at 150 V and low-pressure RF at 60 V. The transitions used in the dMRM analysis are shown in Table 1; the LC and MS parameters used are shown in Table 2.

### 4.7. Data Analysis

Initial data processing was performed using Agilent MassHunter Quantitative Analysis software (v. 10). Post-processing was performed in Excel and GraphPad Prism 9.2.0.

The MRM AUCs were corrected with the matrix response factor (matrix RF) (Appendix A). To determine the matrix RF, the response of the analyte as a standard is needed, as is the response in the sample (the intrinsic contribution, the matrix) and the response of the standard mixture spiked into the matrix. To calculate the matrix RF, the response of the intrinsic contribution in the sample (the matrix) is subtracted from the response of the standard mixture spiked into the matrix, to give the effect of the matrix upon the standard (response of the standard mixture in the matrix). The response of the standard by itself divided by the response of the standard mixture in the matrix gives a matrix response factor. 

To account for matrix effects in biological samples, the response of an analyte in a (biological) sample is multiplied by the corresponding matrix RF value to give the “real” response of the analyte (i.e., to eliminate ionisation enhancement/suppression effects on an analyte). This matrix RF-corrected response can subsequently be used to calculate analyte concentration from a standard curve trendline equation (y = Mx + C). Standard curves are generated by the injection of increasing concentrations of the BLM, calculating the AUC at each concentration, and plotting AUC values against the corresponding concentration.

Statistical analysis of the two sample (or unpaired) *t*-test is used to validate difference between no treatment (NT) and treated with IFNα (+IFNα) for 36 h. A *p*-value less than 0.05 (***) indicates that the results have highly significant differences. 

### 4.8. Method Validation

The LLOD was calculated as a S/N ratio of >3; the lower limit of quantification (LLOQ) was a S/N ratio of >10. Intraday precision was calculated using three replicates of two concentrations over the course of one day and the results are reported as CV % between replicates of one concentration. To monitor instrument performance over time and check that batches that spanned interday analysis were consistent (quality control, QC), the BLM (1 pmol/L on-column) was routinely injected. If the total sample analysis time was over multiple days, the samples were briefly vortexed at the start of each day to avoid precipitation.

## Figures and Tables

**Figure 1 ijms-24-15513-f001:**
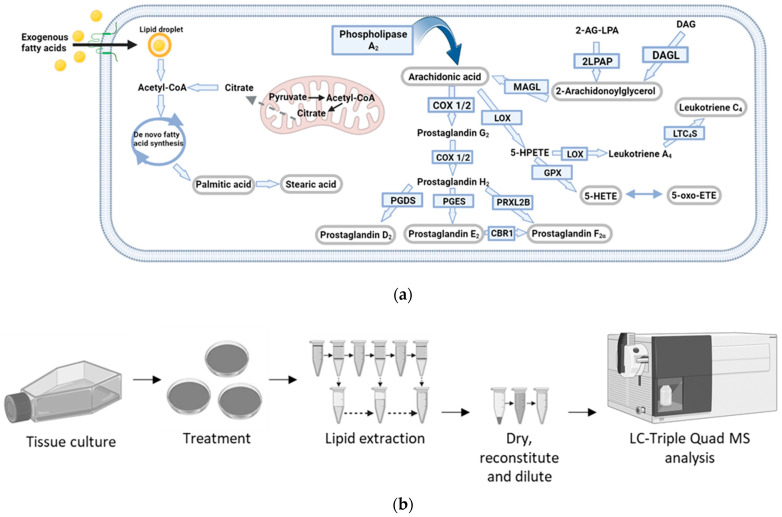
Eicosanoid pathway and methodology workflow. (**a**) Metabolic pathways of selected bioactive lipids. Converting enzymes are in boxes and the lipids in this study are circled in grey. De novo lipogenesis leading to fatty acid synthesis, which results in the elongation of fatty acids, produces Palmitic acid and Stearic acid. Exogenous fatty acid uptake is carried out via transmembrane transporters. (**b**) Optimised experimental workflow for the analysis of eicosanoids and fatty acids was conducted using ultra-high-performance liquid chromatography-dMRM-MS (UPLC-dMRM-MS). Cells were cultured in IMDM media containing 10% foetal bovine serum (FBS) and grown until 60–70% confluent. Cells were then treated with 1000 U/mL of IFNα-2b for 36 h, reaching ~80% confluence. The surrounding media (the cell supernatant) was collected from the plates and cells were lysed. Lipid extraction from cells and cell supernatants was performed with Methanol/Methyl tert-butyl ether/chloroform (MeOH/MTBE/CHCl_3_) and acetonitrile (ACN), respectively. Lipid extracts were subsequently dried and reconstituted in 50% (*v*/*v*) MeOH (in H_2_O) prior to LC-MS analysis.

**Figure 2 ijms-24-15513-f002:**
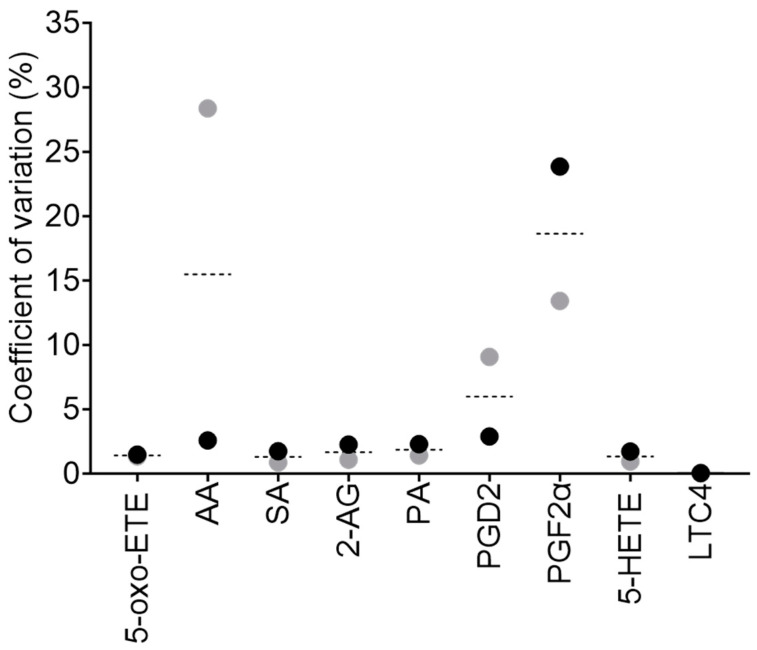
Method reproducibility (intraday analysis). Lipid standards were combined and analysed in technical triplicate at two concentrations, 0.3 pmol (grey dots) and 3 pmol (black dots) on-column, respectively. The CV between technical triplicates was calculated (as %) for each standard, as a mean of the two concentrations (dashed line) or individually (dots).

**Figure 3 ijms-24-15513-f003:**
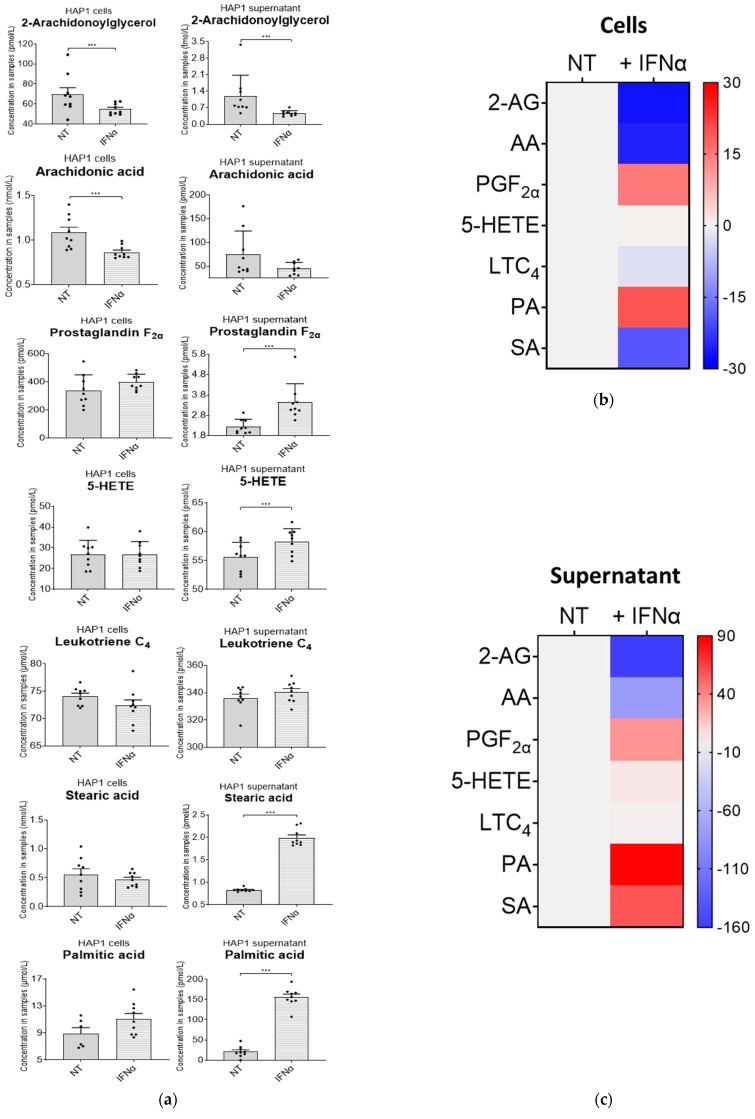
Interferon alpha modulates eicosanoid and fatty acid pathways. (**a**) Analysis of bioactive lipids in HAP1 cells and cell supernatants after 36 h treatment with IFN-I. (**a**) Graphs represent the mean with the standard error of the mean (SEM). Cell results are the graphs on the left; the corresponding cell supernatant is on the right. Grey bars are for the not-treated sample (NT) and grey bars with a white stripe pattern are the results for the IFN-I-treated samples. The statistical analysis is the two sample (or unpaired) *t*-test. A *p*-value less than 0.05 (***) suggests that the results have highly significant differences. If there is no significance difference, it is not labelled. (**b**) Mean percentage change of bioactive lipids in HAP1 cells, not-treated (NT) and treated with IFNα for 36 h (+IFNα). (**c**) Cell supernatant from HAP1 cells, not-treated (NT) and treated with IFNα for 36 h (+IFNα). Heat maps display normalised lipid abundances relative to the mean concentration of the non-treated samples (NT), which is represented by the grey bars. For IFN treatment, blue results are for a decrease in concentration, in percentage, grey is for no change and red is for an increase in concentration in percentage. (**d**) Bioactive lipid profile after 36 h treatment with IFN-I. Converting enzymes are in boxes and the lipids in this study are circled. Lipids circled in grey have no change compared to non-treated cells; lipids circled in blue have a decreased concentration compared to non-treated cells; and lipids circled in red have an increased concentration compared to non-treated cells. Lipids discovered in the cell supernatant are outside of the cell lipid membrane. Lipids with a significant difference between non-treated and treated (a *p*-value of less than 0.05) are indicated by a yellow star.

**Table 1 ijms-24-15513-t001:** Optimised transitions for LC-dMRM-MS. Lipid names, dMRM parameters and lower limits of quantification (LLOQ) are listed for all eicosanoid and fatty acid standards analysed. The CE used for the precursor-to-product ion transitions are indicated in electron volts (eV).

Lipid	Formula	Molecular Weight (g/mol)	Retention Time (min)	Precursor Ion (*m*/*z*)	Precursor Ion Type	Product ion *m*/*z* (Qualifiers)	Product CE (Voltage)	Product Ion *m*/*z* (Quantifier)	Quantifier CE (Voltage)	LLOQ (on Column Concentration)
5-oxo-ETE	C_20_H_30_O_3_	318.22	2.5	319.2	[M+H]+	91.1 43.1 55.1	56 48 68	189	12	500 amol/L
Aracidonic acid	C_20_H_32_O_2_	304.24	3.8	305.25	[M+H]+	58.3 92.1 65.2	28 44 80	91.1	32	20 fmol/L
2-Arachidonoylglycerol	C_23_H_38_O_4_	378.30	7.4	379.3	[M+H]+	91.1 67.2 79.1	72 64 68	287.2	16	100 amol/L
Palmitic acid	C_16_H_32_O_2_	256.43	8.3	257.25	[M+H]+	43.2 57.2 55.2	36 16 48	41.2	68	5 fmol/L
Prostglandin D_2_/E_2_	C_20_H_32_O_5_	350.22	8.3	391.2	[M+K]+	105 63.1 271	24 44 4	312.8	0	5 fmol/L
Stearic acid	C_18_H_36_O_2_	248.27	10.3	285.28	[M+H]+	57.3 120.9 41.2	20 8 72	43.3	36	100 zmol/L
Prostglandin F_2α_	C_20_H_34_O_5_	354.24	10.3	377.2	[M+Na]+	57 43.3 342.2	48 64 12	360.3	4	5 amol/L
5-HETE	C_20_H_32_O_3_	320.24	11.8	338.30	[M+NH_4_]+/[M+H_2_O]+	55.5 41.3 203.3	56 76 20	43.2	52	500 amol/L
Leukotriene C_4_	C_30_H_47_N_3_O_9_S	625.3	14.9	664.26	[M+K]+	57.2 629.5 125	72 24 44	496.3	36	500 amol/L

**Table 2 ijms-24-15513-t002:** Chromatography and mass spectrometry parameters. This table represents the instrumentation settings used in the analysis method and the parameters applied. Settings are selected for the most favourable separation and ionisation of eicosanoids and fatty acids.

Chromatography Setting	Parameter
Mobile phase composition	A = 2% IPA, 5mM Ammonium Acetate B = 100% IPA, 5mM Ammonium Acetate
Mobile phase flow rate (mL/min)	0.21
C18 column temperature (°C)	40
**Source setting**	**Parameter**
Gas temperature (°C)	280
Gas flow (L/min)	14
Nebulizer (psi)	20
Sheath gas temperature (°C)	250
Sheath gas flow (L/min)	11
Capillary: positive and negative polarity (V)	3000
Nozzle:positive and negative polarity (V)	1500
**iFunnel setting**	**Parameter**
High Pressure RF: positive polarity (V)	150
Low pressure RF: positive polarity (V)	60

## Data Availability

Data have been submitted to the public repository metabolomeXchange (Metabolights), reference MTBLS7875.

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
