# Peer review of "Targeted Mass Spectrometry Reveals Interferon-Dependent Eicosanoid and Fatty Acid Alterations in Chronic Myeloid Leukaemia"

_ijms, 2023, doi:10.3390/ijms242115513_

Round 1

Reviewer 1 Report (Previous Reviewer 1)

This paper presents an optimized and sensitive LC-dMRM-MS method for the analysis of eicosanoids and fatty acids in biological samples, and applies it to a cellular model of chronic myeloid leukaemia (CML) treated with interferon alpha 2 (IFN-I).

The paper reveals the effects of IFN-I on the lipid profile of the cells and suggests some lipids may have a role in the inflammatory state of CML tumours.

The paper is well-written, informative, and novel. However, some minor revisions are suggested to improve the clarity and quality of the paper.

The authors should provide more details on how they selected the nine lipids of interest for this study. What are the criteria and rationale for choosing these lipids? How representative are they of the bioactive lipidome in CML?

The authors should explain why they used HAP1 cells as a model for CML. How similar or different are they from primary CML cells or other CML cell lines? How do they reflect the heterogeneity and diversity of CML patients?

The authors should consider moving some figures to supplementary items to reduce the length of the paper. For example, Figure 2, Figure 4, Figure 5b, and Figure 6 could be moved to supplementary items, as they are not essential for understanding the main results and conclusions of the paper.

Author Response

Reviewer 2 Report (New Reviewer)

Dear Authors,

thank you for the very nice article. After reading I have no any considerable issues to be corrected.  The reviewed article represents a good example of a high quality publication with clearly presented experimental data accompanied by logical explanation.  In mine point of view no further improvements are needed and the article can be accepted in a present form.

The only thing that caught my attention was that authors achieved extremely low LLOQs (Tab. 1) for example zmol/L it is 10-21! mol/L...frankly speaking in my practice I have never seen such low LLOQ. Of course, one could, request the initial data to trace if this is true.

Sincerely

Author Response

This manuscript is a resubmission of an earlier submission. The following is a list of the peer review reports and author responses from that submission.

Round 1

Reviewer 1 Report

The paper presents an interesting and novel study on the effect of interferon alpha on the lipid metabolism and inflammation in chronic myeloid leukemia cells. The authors used a mass spectrometry-based approach to profile eicosanoids and fatty acids in HAP1 cells, a near-haploid human cell line derived from a leukemia patient. The paper reports significant changes in several lipids, such as 2-AG, AA, PGF2α, 5-HETE, PA, and SA, in response to interferon alpha treatment. The paper also discusses the potential roles of these lipids in the inflammatory state of tumors and the immune response to interferon alpha.

The paper is well-written and organized, and the methods are clearly described. However, I have some comments and suggestions that could improve the quality and clarity of the paper.

The authors did not detect any changes in LTC4 or 5-oxo-ETE concentrations. These lipids are known to be involved in inflammation and leukocyte activation. How do the authors explain this result? Is it possible that these lipids are rapidly metabolized or exported by the cells? Do the authors have any data on the expression or activity of enzymes involved in these pathways, such as LTC4 synthase or 5-oxo-ETE receptor?

The authors did not detect PGD2 or PGE2 in the samples. These lipids are also important mediators of inflammation and immune regulation. How do the authors account for this finding? Is it due to the low sensitivity of the mass spectrometry method or the lack of expression or activity of enzymes involved in these pathways, such as cyclooxygenase or prostaglandin synthases?

The authors used HAP1 cells as a model for chronic myeloid leukemia. How representative are these cells of the disease? Do they have any genetic or phenotypic similarities or differences with primary chronic myeloid leukemia cells? How do they respond to other treatments, such as tyrosine kinase inhibitors or chemotherapy? It would be helpful if the authors could provide some background information and justification for using this cell line.

The authors used interferon alpha 2 as a representative of interferon alpha family. How does this subtype differ from other interferon alpha subtypes in terms of structure, function, and receptor binding? Are there any studies that compare the effects of different interferon alpha subtypes on lipid metabolism and inflammation in chronic myeloid leukemia cells? It would be interesting to see if there are any subtype-specific effects or interactions.

Reviewer 2 Report

A comprehensive assessment of the manuscript in question is provided below. The paper delineates the advancement of an LC-MS/MS technique for evaluating bioactive lipids in cell cultures. The analytical procedure is introduced with initial cell lysis, followed by LLE extraction. Subsequently, the analytes undergo analysis using the dMRM acquisition mode. The method was subjected to validation and subsequent testing with real samples.

Upon careful examination of the article, it is evident that, from a strictly analytical and methodological perspective within the mass spectrometry domain, it lacks innovation. Specifically, the windowed MRM mode is widely employed in LC-MS/MS and is typically utilized when the quantity of transitions could negatively impact scan duration, potentially leading to decreased method reproducibility. Consequently, this technique is commonplace and cannot be considered groundbreaking within the analytical realm. Additionally, the chosen extraction technique offers no distinct advantages over those detailed in existing literature; it aligns with conventional lipidomics practices.

Furthermore, the method's development and validation procedures necessitate substantial revision by the authors. Given these reasons, I am inclined to believe that the manuscript may not meet the criteria for publication.

Further comments are provided below:

1.      Introduction:

a.      The primary objective of the article remains unclear. It is challenging to discern whether the paper aims to establish a novel analytical method or to enhance understanding of the roles of these molecules within the selected samples. Regardless, the introduction lacks a solid grounding in relevant literature and would benefit from substantial revision.

2.      Results:

a.      The author's rationale behind selecting target compounds should be supported with appropriate references.

b.      The term "optimized" must be used judiciously, and an experimental design is essential to validate its application throughout the manuscript.

c.      Considering the potential presence of numerous isomers in biological matrices, the use of area ratio of transitions is recommended. Sole reliance on fragment analysis might prove insufficient due to potential variations in the ratio of fragments among isomers. Additionally, this approach could help reduce transition numbers, obviating the need for dMRM acquisition mode.

d.      The statement "Interestingly, it was not possible to chromatographically separate the regio-isomer species PGD2 and PGE2 reliably or sufficiently enough to determine their individual concentrations, with this particular methodology" underscores a notable limitation in the method's effectiveness.

e.      The use of CH3COONH4 in the mobile phase prompts inquiry into the consideration of [M+NH4]+ adduct.

f.       The comprehensive nature of the matrix effect assessment appears excessive. Consultation of established validation guidelines is recommended.

g.      The presentation of the matrix effect deviates from conventional approaches and demonstrates substantial variability even within the same matrix type, sometimes up to a 100-fold difference. Consideration should be given to incorporating a clean-up step or evaluating the implementation of internal standards. The significant matrix effect variability contradicts the high method reproducibility observed.

3.      Validation:

a.      The validation process falls short due to the absence of linearity and accuracy tests. Furthermore, the chosen levels for recovery, matrix effect, and accuracy assessments do not align with the concentration range demonstrated by the linear trend in the supplementary materials. To ensure comprehensive validation, these levels should encompass the full linear range of the method.

b.      Although the authors assert the method's robustness, no dedicated tests have been conducted to confirm this claim.
